# Effectiveness of a positive deviance approach to improve appropriate feeding and nutritional status in South West Region, Ethiopia: A study protocol for a cluster randomized control trial

**Abraham Tamirat Gizaw** [1]*, **Pradeep Sopory**[2], **Morankar N. Sudhakar**[1]

**1** Faculty of Public Health, Department of Health, Behavior, and Society, Institute of Health, Jimma University, Jimma, Ethiopia, **2** Department of Communication, Wayne State University, Detroit, Michigan, United States of America

\* abraham.tamirat@ju.edu.et, abrishntamirat@gmail.com

## Abstract

### Background

Non-optimal infant and young child feeding practices (IYCFP) are linked to malnutrition and infant mortality in poor countries, notably in Ethiopia. The majority of growth stalls occur within the first two years of life; hence, there is a need to discover interventions that enhance appropriate IYCFP for improving nutritional outcomes during this critical period. Using the experience of mothers who have come up with solutions to their IYCFP problems to educate others, is a potential pathway to initiate and sustain behavioral changes in resource-limited areas. However, such interventions are not widely implemented in Ethiopia.

### Objective

This study aims to assess the effectiveness of a positive deviance approach (PDA) to improve appropriate feeding and nutritional status in South West region, Ethiopia.

### Methods

A cluster randomized controlled trial will be conducted to compare the effect of positive deviant intervention versus routine health education. The intervention will be provided by positive deviant mothers based on uncommon practices that potentially benefit IYCFP will be identified. Training of the trainers manual on counselling and supporting non-positive deviant mothers based on the uncommon practices identified and WHO IYCFP guidelines will be provided for facilitating change. Culturally appropriate and scientifically acceptable operational packages of information will be developed. A total of 516 mothers will be recruited from 36 zones. The intervention arm will receive 12 demonstration sessions in groups and every 15th day home visit by positive deviant mothers. Data will be entered into epi data version 3.1 and analyzed using STATA version 16.0. All analyses will be done as intention-to-

**Data Availability Statement:** No datasets were generated or analysed during the current study. All

relevant data from this study will be made available upon study.

**Funding:** The authors received no specific funding for this work.

**Competing interests:** The authors have declared that no competing interests exist.

treat. We will fit the mixed effects linear regression model for continuous outcomes and mixed effects linear probability model for the binary outcomes in the study zone as a random intercept to estimate study arm difference (intervention vs. routine health education) adjusted for the baseline values of the outcome and additional relevant covariates.

## Discussion

We expect that the trial will generate findings informing IYCFP and nutritional policies and practices in Ethiopia.

## Trial registration

This trial was registered at ClinicalTrials.gov as PACTR202108880303760, 30/8/2021.

## Introduction

The World Health Organization (WHO) recommends exclusive breastfeeding for the first six months of a baby's life, followed by the introduction of nutritious complementary foods while continuing breastfeeding up to two years of age or beyond [1]. Infant and young child feeding practices (IYCFP) have the largest potential influence on child survival of any established preventive health and nutrition intervention framework. Malnutrition in children mostly occurs during the first two years of life due to a high demand for nutrients to sustain rapid growth and development; hence, the first two years of life are seen as a key window of opportunity to reduce malnutrition [2]. The period between 6 and 24 months of age is a time of nutritional vulnerability because nutrients, particularly micronutrients, and energy supplied only from breast milk will not be adequate to fulfil the child's needs. Therefore, ensuring appropriate nutrition between the ages of 6 and 24 months is vital for the health and growth of the child [3–6].

Infant mortality and malnutrition in children are the most sensitive indicators of a country's growth and development. IYCFP is a highly concerned global public health issue for its extensive role in child development, growth, and survival [7]. In 2010, approximately 104 million children under the age of five were classified as underweight and 171 million were considered stunted globally. Out of the stunted children, about 90 percent were living in 36 countries. The age group most vulnerable to malnutrition is children under the age of two. Malnutrition is responsible for approximately 5.6 million child deaths each year, with severe malnutrition accounting for around 1.5 million of these fatalities. Undernutrition is linked to 45 percent of all infant deaths, totalling 2.7 million worldwide annually [8].

In Ghana, only around 12.0% of young children are fed with infant feeding bottles [9]. While undernutrition (low weight for age) is widely recognized, the relevance of acute malnutrition is rarely, if ever, emphasized. This is a significant omission; acute malnutrition is very prevalent with high rates of death and morbidity that need specific treatment and preventive measures [10]. Maternal low knowledge, negative attitudes, low self-efficacy, and cultural influences and taboos all have a significant impact on mothers' infant-feeding practices and their children's eating patterns [11]. Studies conducted in different parts of the world, such as in Brazil, revealed that knowledge, culture, self-efficacy, and beliefs are strong predictors of IYCFP [12]. Another study conducted in Turkana in Africa, also found that although mothers were aware of the length of exclusive breastfeeding, 85.6 percent reported that EBF should last

6 months, and most were unaware of the need for continuing breastfeeding beyond the age of 24 months [13].

Sub-Saharan Africa (SSA) has one of the lowest breastfeeding rates, with 37% of infants aged less than six months being exclusively breastfed. Lower proportions of infant feeding practices have been recorded in several SSA nations, where diarrhoea is still a major cause of morbidity and mortality. Lower socioeconomic levels, home childbirth, culture, and inadequate implementation and monitoring of programs are all plausible causes for inadequate exclusive breastfeeding practices in Africa [14]. The Ethiopian government developed and implemented the IYCFP guideline in 2004 to improve feeding practices and made different agreements to reduce infant mortality that results from malnutrition, including the Sekota agreement that aimed for a 1,000-day nutrition service program [15]. However, children's malnutrition remains a significant public health challenge in Ethiopia.

According to the available research, in Ethiopia, poor dietary practices among children reported being malnourished ranged from 15.2–35.5%. According to the research conducted in Shashemene, 215 (65.7 percent) children aged 6 to 24 months began solid, semi-solid, and soft meals between the ages of 6 and 8 months. Only 128 (39.1%) of infants aged 6 to 24 months fulfilled the criteria for minimum dietary diversity (MDD), which is the consumption of four or more food groups from the seven food groups [16]. The percentage of stunted and underweight children is greater in rural areas than in urban ones, reflecting poor childcare practices in rural areas. Eighty-five percent of children scored poorly on dietary diversity [17–19]. According to a study conducted in the Jimma zone on complementary feeding practices, the majority (88.9%) of the children were exclusively breastfed, and 75.6% were breastfed up to the age of two years [3].

The term positive deviance (PD) describes the performance (regarding health, growth, and development) of certain children in the community in terms of positive social, behavioral, and physiological adaptability to nutritional stress [20]. The approach has been used by maternal and child health programs since the 1970s as a way of addressing childhood malnutrition by learning from and scaling up what is working rather than focusing on what is not working [21].

Currently, positive deviance is increasingly being used in international development activities to permit the utilization of proven local solutions in problem-solving and as a means of generating local involvement in addressing these problems [22]. Individuals within a community are identified who demonstrate exceptional behaviors (uncommon behavior) or practices that enable them to get better results than their neighbors with the same resources [23,24]. Positive outcomes include improved dietary intake, fast weight gain for severely malnourished children, uptake of exclusive breastfeeding practices, and decreased morbidity in intervention communities compared to non-intervention communities [25,26]. There is also ample evidence across the globe that utilized PDA brought positive nutritional outcomes. For instance, a study conducted in Mozambique and Burundi PDA applied in tackling undernutrition was evidence of the application and contribution of PDA in different areas of public health [27]. Therefore, this study aimed to evaluate the effectiveness of PDA in improving IYCF knowledge, attitude, self-efficacy, and nutrition status, in the rural setting of South West, Ethiopia.

## Materials and methods

### Ethical approval

The protocol was developed in collaboration with the West Omo Zone and Maji Woreda Health Office. Ethical approval (Ref no: IHRPG/938/2020) was obtained from Jimma University, Institute of Health Research and Postgraduate Office, Jimma, Ethiopia.

## Design

A cluster-randomized controlled single-blinded parallel-group, two arms trial with a 1:1 allocation ratio was designed to evaluate the effectiveness of the positive deviant approach (PDA) provided for mothers to improve mothers' IYCF knowledge, attitude, self-efficacy and children's nutritional status. This study design was chosen to avoid contamination among treatment groups. Clusters are zones, i.e., small administrative units found in Maji Woreda, West Omo Zone

Formative research will also be conducted to understand the barriers to IYCFP using quantitative methods as baseline studies. Furthermore, the socio-cultural and contextual factors influencing IYCFP will be explored.

## Setting

This study will be conducted in the Maji district, West Omo zone, in Southern Ethiopia. Maji is one of the districts in the South West region. Maji is bounded on the south by the Kibish River, which divides it from South Sudan, on the west by Surma, on the northwest by Bero, on the north by Meinit Shasha, and on the east by the Omo River, which separates it from the Debub Omo Zone. There are 117 kebeles in the zone, with a population of 1,272,943. The Maji district has a total of 22 kebeles. In the Maji district, there is a district hospital, four health centers, and 22 health posts. Maji district is semi-pastoralist, with the bulk of the people relying on traditional rain-fed agriculture and rearing cattle. Maji district has a population of 230,777 people and is located 817 kilometers from Addis Ababa, Ethiopia [28]. The district is split into kebeles, which are the lowest administrative units, and each kebele is divided into four small zones.

## Eligibility criteria for clusters

Out of the 88 zones found in Maji Woreda, 36 non-adjacent clusters that are geographically accessible will be selected purposively for the study, resulting in 18 interventions and 18 controls.

## Eligibility criteria for the participants

The study will comprise Maji district mothers and their infants and young children (IYC), aged 0–24 months. The non-positive deviant mothers will be selected based on the following inclusion criteria: mothers with index child aged 0–24 months living in the selected clusters with no plan to move away during the intervention period, capable of giving informed consent, and willing to be visited by supervisors and data collectors. Exclusion criteria; mothers with severe psychiatric illness which will interfere with consent, the child Height-for-age Z (HAZ) < -3 Z score (severe stunting), and children with severe illness or clinical complications, which will potentially influence the study outcomes. However, such children will be referred to the nearby health institution for better care. The participants will be recruited from April 15 to October 19, 2022

## Sample size determination

The sample size was calculated using statcalc (STATA software version-16) with the following assumptions: to detect an increase in appropriate feeding from 7% to 14% [29], with 95% CIs and 80% power, assuming an intra-class correlation coefficient of 0.03 [30] equal to the Ethiopian study for a cluster size of 12, it was calculated that 36 clusters needed. This gave a sample size of 215. Then it was multiplied by the design effect of 2 and allowing for 20% of the sample

size for loss to follow-up, the total sample size was 516 mothers (258 from the intervention arm and 258 from the control arm).

## Sampling and randomization procedures

Zones in the kebeles will form the unit of randomization for the trials, while mothers within the zones will form the unit of observation. From seven woredas in the West Omo zone, Maji Woreda was selected purposively. After identifying and listing the 88 zones found in Maji Woreda, 36 non-adjacent zones will be selected porpusively. Then, eligible mothers will be identified from the selected zones using the health extension worker's logbook before the zones are randomized either into intervention or control groups. Simple randomization with 1:1 allocation will be used to randomize the zones to either the control or the intervention groups. First, the 36 zones will be listed alphabetically, and then a list of random numbers will be generated in MS Excel 2016, and the generated list will be fixed by copying them as "values "next to the alphabetic list of zones. These will be arranged in ascending order according to the generated random numbers. Finally, the first 18 zones will be selected as intervention clusters and the last 18 as control clusters. Interviewers collecting outcome data are blind to the intervention assignment. The data collectors will be recruited out of the study, and the clusters will be coded (0–136) and the households will be labeled (001–516) in such a manner that the research team can identify which clusters and households are in the intervention and which are in the control group. Due to the nature of the intervention, it is not possible to blind mothers. However, all mothers, health extension workers, women, the health army, and community volunteers are blind to the study hypothesis. The consent for data collection includes a general description of the overall aims of the study. Fig 1 shows selection process in details.

## Data collection

The data will be collected on socio-demographic knowledge, attitude, and self-efficacy of breastfeeding and complementary feeding. The data will be collected by an interviewer-administered questionnaire in the Amharic language. Three BSc nurses and ten diploma holders in the health center will be recruited as supervisors and data collectors, respectively. The supervisors will supervise and coordinate the data collection process.

## Intervention

**Positive deviance approach (PDA).** The researchers, village leaders,women representatives, health development army leaders (HDALs), and health extension workers (HEWs) will cooperatively design artifacts, workflow, and work environments; and argue iteratively so that they develop and refine their understanding of the activities. With the use of this approach, researchers are able to comprehend tacit and implicit information, which is what mothers are aware of but are unable to express. This knowledge is holistic and not limited and systematized. The mothers have experience, but their wisdom is too deep to be fully expressed. These will be clarified by the positive deviance inquiry since knowledge is a requirement of a particular setting.researchers, village leaders, women, representatives, HDALs, and HEWs will identify positive deviants and practices using six steps [24,25].

**Step 1:** Defining the problem: We will work together with the team in defining the problem, or why mothers inappropriately feed infants and young children (IYC). In essence, the problem is related to breastfeeding and complementary feeding. The main agenda for discussion will be "What are the problems related to IYCFP and why? Emphasizing breastfeeding and complementary feeding. The principal investigator will be the facilitator of the overall qualitative enquiry to ensure whether it is part of the intended intervention or not.

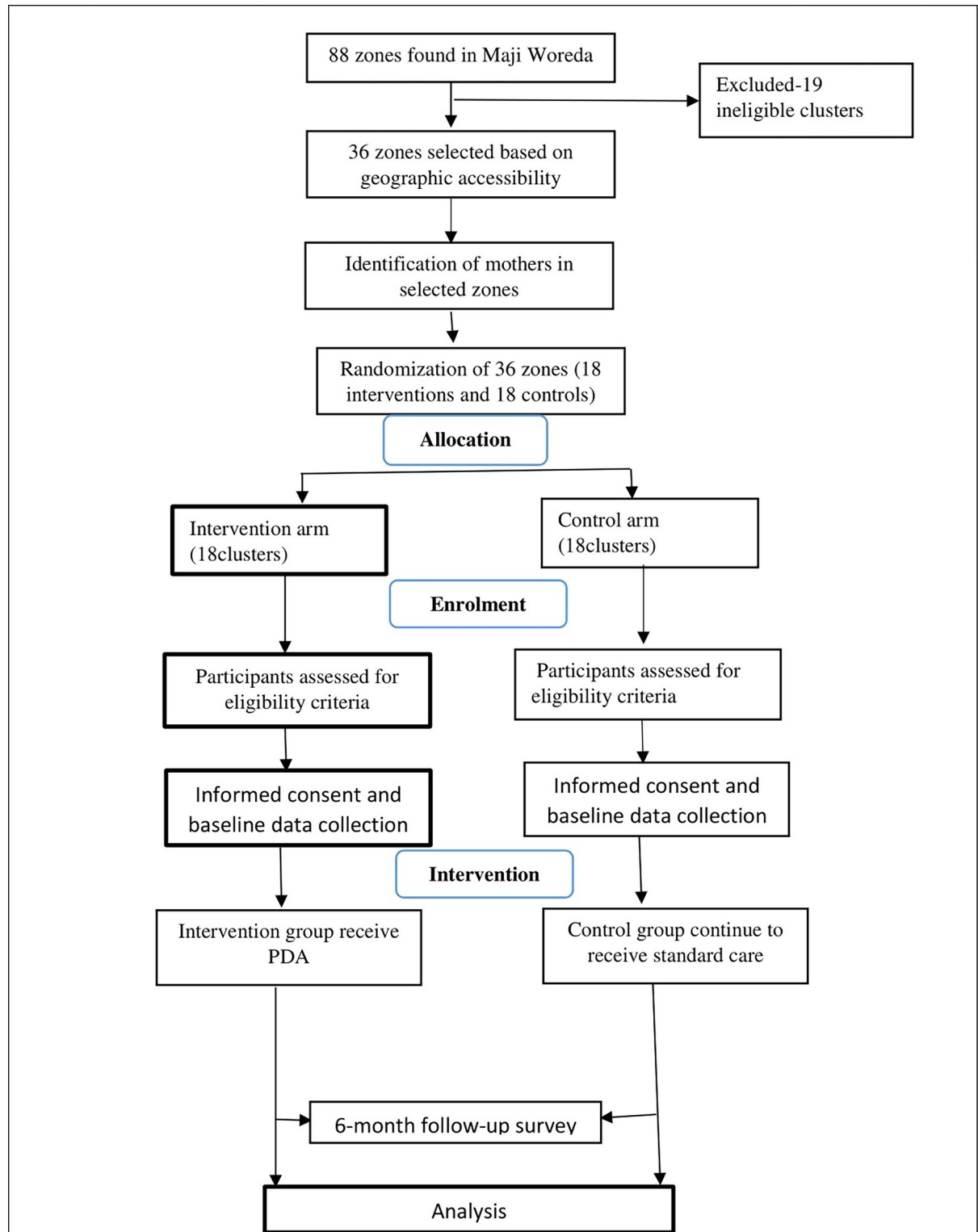

**Fig 1. Flow diagram of study population.**

**Step 2:** Determining the presence of positive deviant mothers (PDM): after defining the existing problems related to IYCFP, the village leaders, women representatives, HDALs, and HEWs will support the researchers in identifying PDM in the community. We will try to investigate whether some PDM exist and have lived experience to overcome their problems that will be generated during group discussion with the team. After confirming their presence, we will develop a list of criteria to select PDM from the community by shortlisting, completing the checklist/criteria, and interviewing from the final list. The following criteria will be used to assess the candidate PDM; poor family, a family that is representative of geographical and social groups living in the village, belonging to the community, head of the household, same occupation as the majority of villagers, previous best experiences of IYCFP (if multi-para, for the last child) regardless of any challenges, have good knowledge, positive attitude, confidence, good skills of breastfeeding or/and complementary feeding approved through interviews and demonstration, willingness or motivation, and good communication skills demonstrated during interviews and full attendance at the training will be used to select the PDM. Furthermore, a PDM is a mother who goes against traditional norms and expectations to create positive change and improve the well-being of her children. These mothers take unconventional approaches to parenting, often challenging societal norms and beliefs to ensure the best outcomes for their children. PDM are those mothers who are exclusively breastfed for 6 months and introduced complementary foods at the age of 6 months.

**Step 3:** Discovering uncommon practices or behaviors: the team will try to discover the best practices that are uncommon and scientifically sound as per the guidlline of IYCFP. After discovering uncommon practices: the project team, in discussion with PDM, will determine the practice or behaviors that allow them to appropriately feed their child. Therefore, the study will focus on tracing intentional and unintentional practices that enable them to achieve better IYCFP. Hence, a mother who overcomes the identified challenges to appropriate IYCFP will be considered as uncommon practice. A range of activities will also be carried out simultaneously, such as developing IEC materials and messages, promoting IYCFP based on the local context.

**Step 4:** after identifying successful practices, HDALs, HEWs, and PDM will decide what strategies they will like to follow and design activities to enable others to practice uncommon practices. Finally, strategies will be added to the adopted strategies for the intervention.

**Step 5:** the project will be monitored by the researchers, supervisors, HDALS, and HEWs. Process evaluation will be carried out will be carried out through a review meeting. Activities will be monitored to see the implementation is taking place as planned or not to take remedial actions. To ensure this, process, evaluation will be carried out at the beginning, midline, and endline.

**Step 6:** scaling up depending on the success of the research scale-up will be considered as a sustainability plan. It is expected that the community after observing the effectiveness, they will intend to join PD teams for the better outcome. After the completion of the study, positive outcomes will be recommended for scale-up for other kebeles, woreda zonal, regional, and nationally at large. In the process of scale-up of this research, finding notably directly benefits infants and young children.

## Intervention activities

**For intervention groups.**   A non-positive deviant mother related to IYCFP refers to a mother who exhibits inadequate feeding practices such as initiated breast feeding late, weaned their children off of breast milk early, and too early or lately introduced complementary feeding. Generally, these mothers deviate from recommended guidelines IYCFP. Once the non-

positive deviant mothers will be identified, then the intervention with different modalities will be carried out. Nutrition education sessions including demonistration will be provided for the selected non-positive deviant mothers in the intervention for 12 continuous days in groups in their nearby setting in the community. The intervention is composed of the following elements: a) breastfeeding education to raise knowledge, attitudes, and self-efficacy toward breastfeeding, b) complementary feeding education to raise knowledge, attitude and self-effi-cay toward complementary feeding, c) counseling on how to increase consistency, quantity, and frequency of foods, using locally available foods, d) practical demonstration how to cook locally available food items, counseling, and e) hygiene support. Moreover, to support the lived experiences with the scientific way, the intervention arm will receive nutrition education with standard manuals prepared in the local language. During each visit, PDM will cover the details of the importance of breastfeeding, complementary feeding, and feeding an ill child. The discussion will combine the use of information, education and communication (IEC) materials and practical demonstration on appropriate IYCFP.

Mothers will be encouraged to ask any questions related to the topic discussed. PDM will use culturally appropriate language in the form of a poster to illustrate the new information (eg., correct and incorrect breastfeeding, preparation of enriched flour, appropriate consistency (thickness) and inappropriate consistency of complimentary food, the importance of significant others support) and the benefit of applying the recommended IYCFP (pictures of the babies who were appropriately fed versus those who were not). PDM will visit non-positive deviant mothers every 15th day, with each mother visiting twice a month for education and demonstration. Education sessions and demonstrations will take from 45–60 minutes for each mother, all the necessary logistics for the intervention will be supplied by the reaearch team conducting the trials. During these sessions, PDM will also provide IEC materials as reminder to recommended IYCFP. During each visit, mothers will be observed breastfeeding, ensuring appropriate feeding and preparation for complementary feeding, solving any breastfeeding problems, addressing inappropriate feeding, promoting dietary diversity and consistency, and offering hands-on guidance when necessary. They will support and encourage the mothers to follow appropriate IYCFP from 0–24 months. PDM will also promote personal and home hygiene, such as handwashing before feeding, after using the toilet, and when changing babies' diapers.

Every month, the principal investigator will get feedback from positive deviant mother (PDM) to identify if they have faced any challenges such as technical as well as medical problems. The checklist will also confirm the presence of PDM per predetermined schedule (both non-deviant as well as PDM) will sign on it. The feedback will be given and the solution will be sought for both technical and medical issues. If the medical issues will be raised to both the mother and a child, immediate referral to the nearby health centers will be made. The whole trial work plan is presented in Table 1.

**Supervisors.**    Three persons will be involved in the supervision of the PDM. The supervisors' main responsibilities are to provide supportive supervision and monitor the PDM. Supervisory visits will be conducted by the researcher along with the supervisor every fifteen days. PDM will receive feedback on their work from the supervisors during monthly supervision meetings.

**For control arms.**    The control arm will receive the usual routine services provided by the health extension workers in their kebeles and zones.

**Baseline data collection.**    Both quantitative and qualitative studies will be employed for the baseline data collection in the study area. Formative studies will be conducted to assess the level of knowledge, attitude, and self-efficacy of the mothers as a baseline using quantitative methods. To understand the contextual factors as well as barriers towards IYCFP a qualitative method will be employed.

**Table 1. The trial work plan.**

| Activities | Study period | | | |
|---|---|---|---|---|
| | *-t₁* | **Baseline (t0)** | *Midline at month 3 (t1)* | *End line at month 6 (t2)* |
| Enrolment and allocation | I+C | | | |
| Socio-demographic assessment | | I+C | | |
| Household food insecurity assessment | | I+C | | |
| Water hygiene and sanitation assessment | | I+C | | |
| Cultural beliefs and food taboos | | I+C | | |
| Infant growth | | I+C | I+C | I+C |
| Breastfeeding knowledge, attitude and self-efficacy data | | I+C | I+C | I+C |
| Complementary feeding knowledge, attitude and self-efficacy data | | I+C | I+C | I+C |

I = Intervention groups; C = Control groups; I+C = Activities both in intervention and control groups.

## Study hypotheses

- Compared to the mothers of the control group, IYCF knowledge, attitude and self-efficacy of mothers in intervention group will be significantly improved over the six months of follow-up.

- Compared to the children of the control group, nutritional status in children of the intervention group will be significantly improved over the six months of follow-up.

## Primary objective

This study has two primary objectives:

First, at baseline, midline, and endline, this study will compare the effectiveness PDA versus routine nutrition-related health education provided by health extension workers (community health workers) on mothers' knowledge, attitudes, and self-efficacy towards infant and young child feeding (IYCF). Second, the nutritional status in the control and intervention arms will be compared at baseline and 6-month follow-up (Table 2).

**Table 2. Primary and secondary outcome variables among study participants at baseline, 3 months, and 6 months.**

| Outcome measure | Variables | | Scale | Type | Measure | Analysis |
|---|---|---|---|---|---|---|
| Primary outcomes related to IYCFP (breastfeeding and complementary feeding) | Mothers' knowledge | | Interval | Continuous | Change in knowledge score | *t*-test |
| | Mothers' attitude | | Ratio | Continuous | Change in attitude score | *t*-test |
| | Mothers' self-efficacy | | Ratio | Continuous | Change in self-efficacy scores | *t*-test |
| Primary outcome related to nutritional status | Weight | | Ratio | Continuous | Change in weight | *t*-test |
| | Height/length | | Ratio | Continuous | Change in height | *t*-test |
| | Weight-for-height z-score | | Ratio | Continuous | Change in weight for height | ANOVA[a] |
| Secondary outcomes | Barriers and facilitators of IYCF practices | | - | - | Codes, categories, and emerging themes. | Thematic analysis |
| | Child morbidity status | Diarrhea | Nominal | categorical | % of a child with diarrhea | Risk ratio |
| | | Fever | Nominal | categorical | % of a child with fever | Risk ratio |
| | | Cough | Nominal | categorical | % of a child with cough | Risk ratio |

[a]ANOVA: Analysis of variance.

### Secondary objectives

This study also has one secondary objective, which is to explore barriers towards IYCFP at baseline through a qualitative study to supplement the quantitative study.This will help to understand the cultures and other factors influencing IYCFP in the study area and the child morbidity status will also be assessed through direct interviews of mothers.

## Outcome measurements

### Primary outcomes

**Breastfeeding knowledge, attitude, and self-efficacy** will be measured as the breastfeeding knowledge score, attitude score, and breastfeeding self-efficacy score of mothers with index children from 0–24 months.

   **Complementary feeding knowledge, attitude, and self–efficacy** will be measured as the complementary feeding knowledge score, attitude score, and self-efficacy score of mothers with index children from 0–24 months.

   **Child growth** growth standards will be used to estimate anthropometric status. Weight-for-length z-score (WLZ), length-for-age z-score (LAZ), and weight-for-age z-score (WAZ). Children who have WLZ below- 2 (WLZ $< - 2$) will be considered wasted, those with LAZ below- 2 (LAZ $< - 2$) stunted, and those with WAZ below—2 (WAZ $< - 2$) underweight [1].

### Secondary outcomes

**Barriers to IYCFP** will be explored using in-depth interviews and focus group discussions with mothers of index children (0–24).

   **Child morbidity status** will be determined through direct interviews with the mothers on various health indicators and diseases related to the index child (0–24 months). The mother will be asked if the child had any illness, diarrhea, or cough in the two weeks prior to data collection.

## Data collection tools and techniques

### Breastfeeding measurements

All data collectors will be trained on the content, questionnaire techniques, and measurements; in addition, production and validity exercises will be conducted for the weight and length measurements. One of the requirements for data collectors and supervisors will be knowledge of local culture and field study expertise. A breastfeeding knowledge, attitude, and self-efficacy questionnaire will be developed and adapted using the following validated instruments: Breastfeeding knowledge [31], the Iowa Infant Feeding Attitude Scale (IIFAS) [32], and the short form of breastfeeding self-efficacy scale (BSES-SF) [33].

   Breastfeeding knowledge consists of 17 items to measure the knowledge of the participants about breastfeeding. There are three possible responses for each item (true, false, and I do not know or not sure). Correct responses will be scored as one and zero for other options. Thus, the total scores ranged from 0–17, these items are developed based on a study done among the Chinese mothers in English and translated to Amharic [34,35].

   The Iowa Infant Feeding Attitude Scale (IIFAS) consists of 17 items on a five-point Likert scale, rating maternal attitude towards breastfeeding translated from English will be used. The scales ranged from strongly disagree to strongly agree on each item to indicate attitude towards infant feeding. A sum of scores ranging from 17 to 85, with the higher score reflecting a positive attitude whereas the lower score showing negative attitude. Attitude toward breastfeeding

will be categorized as follows: (1) positive to breastfeeding (IIFAS score 70–85), (2) neutral (IIFAS score 49–69), and (3) positive to formula feeding (IIFAS score 17–48) [36]. The IIFAS is a validated and reliable measure (Cronbach's alpha score range from (0.81–0.86) that evaluates breastfeeding attitudes in different cross-cultural settings [32,36,37]. Approximately half of the questions will be negatively worded (i.e., 1, 2, 4, 6, 8, 10, 11, 14, and 17) [34].

The short form of breastfeeding self-efficacy scale (BSES-SF) is widely used with a variety of population and journals [38,39]. The overall score of the scale will be calculated as the mean score of all items. A higher total score is indicative of a greater level of maternal breastfeeding self-efficacy. BSES-SF consists of 14-items with a five-point Likert scale, developed to measure breastfeeding confidence in Amharic translated from a validated English questionnaires from different studies, which measures the mother's self-efficacy in her ability to breastfeed. All items are preceded by the phrase "I can always "and anchored with a 5-point Likert scale, where 1 indicates not at all confident and 5-indicate always confident. All items will be presented positively and scores will be summed to produce a range from 14 to 70 [40].

## Commentary feeding measurements

A standardized pretested structured interviewer administered questionnaire will be employed for data collection. The adapted questionnaires are from IYCFP indicator parameters [44] and previous studies conducted by different researchers. Complementary feeding knowledge questionnaires are adapted after reviewing different researches conducted in different settings. The followings are different researches that the questionnaire adapted [41,42]. Components of complementary feeding knowledge are: duration of continued breastfeeding, age of start of complementary feeding, and reasons for giving complementary feeds. The knowledge scale has 10 items and consists of both open-ended questions and multiple choice questions. Each question scores 1 for correct answer and 0 for incorrect answers. Total scores will be generated for each participant and computed out of 100%.

Components of the attitude towards complementary feeding consisted of 8 items on a five-point Likert scale, rating maternal attitudes towards complementary feeding translated from English and assess: giving a variety of meals and feeding frequencies. Total scores will be generated for each participant and computed out of 100% [43,44]. Negatively worded items will be inversely coded during analysis.

Complementary feeding self-efficacy consists of 9-items on a five-point Likert scale, developed to measure mothers' confidence towards complementary feeding adapted from a previous studies will be used [45]. Components of self-efficacy towards complementary feeding are: confidence in giving a variety of meals and feeding frequency. Total scores will be generated.

Anthropometry: recumbent length will be measured to the nearest 0.1 cm using a portable wooden infant/child length board with a fixed head and a sliding foot pieces. Infants will be weighed with light clothing to the nearest 10 g using a salter scale. The weighing scale and length board will be placed on a flat surface to ensure correct measurements. Each measurement will be done in duplicate and the mean value calculated. Standardized anthropometric procedures will be observed. Nutritional status indices; height-for-age, weight-for-age, and weight-for-height Z scores will be computed, and will be determined for each child by comparing the child's measurements with the reference values of the child growth standards using ANTHRO software (at baseline and endline in both study groups).

## For qualitative data collection

In-depth interview and focus group discussion guide will be adapted from WHO indicators for assessing IYCFP, consisting of five modules on IYCFP: (a) breastfeeding, (b)

complementary feeding, (c) preparation and storage of foods and drinks, (d) breastfeeding and complementary feeding barriers and (e) cultural norms and gender issues. The interview guide included open-ended questions and a free listing approach to collect perceptions of barriers towards IYCFP [46].

### Child morbidity measurement

The three illnesses will be diagnosed using standardized assessments similar to those used in previous studies. The mother will report any childhood morbidity. The mothers will be asked if their child had any illness, diarrhea, or cough in the two weeks prior to data collection. Mothers of children who have diarrhea will be asked if there is blood in their stools. We will inquire about the child's breathing difficulties if he or she has a cough. An acute respiratory tract infection was defined as persistent coughing or difficulty breathing over the previous two weeks. A diarrhea episode is defined as having at least three loose stools in one day [47].

### Data management

All filled questionnaires will be cheked for completeness by supervisors and questionnaires with missing items will be returned to the data collectors for correction. Mothers who are lost to follow-up will be recorded along with their reasons.The following standard process will be implemented to improve the accuracy of the data entry and coding: double data entry and coding; verification that the data is in the proper format and within an expected range of values; and independent source document verification of a random subset of data to identify missing or apparently erroneous values. In order to ensure confidentiality, information about each zone and personal data of the participants will not be shared with any third party, both during and after the trial.

### Data possessing and analysis

EpiData version 3.1 will be used for double data entry, and Stata version 16 (StataCorp) will be used for consistency checks and statistical analysis. Descriptive statistics will be utilized to assess and summarize sociodemographic, socioeconomic, child health status, child morbidity, and child feeding. The data for continuous variables will be given as a mean and standard deviation, or median and range, whereas categorical variables will be reported as a frequency and percentage. The t-test and analysis of variance will be used to compare group means for primary and secondary outcome variables. The Chi-square test will be used to examine the categorical variables. Child nutrition outcomes will be computed and compared to the WHO growth standards [48].

The results of group comparisons will be reported as a risk ratio for binary outcomes, equivalent to 2-sided 95 percent confidence intervals and related p values. All p values will be scaled to two decimal places, with values less than 0.01 reported as < 0.01. Adjusted analyses utilizing baseline variables will be done using multivariate logistic regression to assess the ongoing effect of important baseline features on outcomes. For time-dependent variables such as morbidity, the Cox's proportional hazard model will be employed. The intention-to-treat analysis will be utilized, and the clustering effect (using the study zone as a random intercept to account for clustering of subjects by zones) will be addressed. All analyses will be done with a 95% confidence interval. The significance level will be assigned at a p-value < 0.05.

For the qualitative data, IDIs and FGDs will be audio-recorded and transcribed verbatim. The principal investigator will check the transcripts against the original recording and field notes for accuracy. Atlas_ti analytical software (version 7.5.18) will be used for coding and analyzing the data. All transcripts will be analyzed inductively with respect to the following phases

of thematic analysis: familiarization with the data, generating initial codes, selection, review, definition, naming of themes, and reporting [49]. The codes will be reviewed and discussed by the researchers and will be then grouped into themes under each general topic area. Lastly, descriptive quotes will be selected to support the themes.

## Data quality assurance

The following strategies will be used to assure the quality of the data;

- The questionnaires will be prepared in English and translated to the local language Amharic and then back to English by experts of the language to keep its consistency,

- Careful selection and training of data collectors and supervisors on the tools and methods of data collection by the principal investigator,

- A Pre-test will be done on 5% of the samples in a community with a similar status to the study community before the actual data collection to check the completeness, consistency, and applicability of the instruments, and will be ratified accordingly,

- Training materials for the intervention clusters will be assessed after conducting training sessions and will be ratified accordingly if needed,

- Standard procedures will be applied in conducting anthropometry,

- Filled questionnaires will be checked for completeness and consistency of information by the supervisors on a daily basis,

- Daily supervision in every step of data collection will be made and

- To enhance blinding, the precise objectives of the study and village allocation to the trial will not be disclosed to data collectors, HDA leaders will not be responsible for data collection, and the data collection schedule will be randomized.

- The Validity and reliability of the instrument will be ensured through external and content validity. Test–retest method will be used to assess the reliability of the instrument.

## Discussion

In the experimental arm, the PDA will be provided concurrently to mothers to enhance appropriate IYCFP and nutritional status. To ensure accessibility and sustainability, the intervention is culturally adapted for implementation in the community context by selecting positive deviant mothers who solved their problems with locally accessible food items and brought about major changes in child feeding. Essentially, complementary feeding will be based on locally available and consumed food items in the community. It is expected that mothers will perceive child undernutrition as an important health concern and will be motivated to practice recommended IYCFP. This study will improve the knowledge, attitude, and self-efficacy of mothers towards breastfeeding and complementary feeding, and nutritional status. The practical session of the intervention includes a demonstration of how to cook and feed a child will improve skill acquisition. We predict that our PDA will improve appropriate IYCFP from 7% to 14%, nutritional status, knowledge, attitude, self-efficac, and other cultural and gender issues influencing IYCFP as compared with routine nutritional education provided by health extension workers (community health workers). Finally, the findings will be scaled up to other regions of Ethiopia.

This study has its own strengths and limitations. The strengths of this study are; first, it promotes equity by drawing on the knowledge of the disadvantaged regarding healthy behavior and offering solutions that are affordable for those who face comparable socioeconomic challenges; second, it introduces a general approach to solving local problems. However, PDA has the following limitations; it requires discovering uncommon behaviors/ practices that can be rare in the community and scale-up requires many people with skills in community mobilisation, participatory research, and positive deviance, which could limit acceptance.

## Supporting information

**S1 File. SPIRIT checklist.**
(DOC)

**S2 File. Protocol.**
(DOCX)

**S3 File. WHO trial registration data set PDI.**
(DOCX)

**S4 File. Model consent form.**
(DOCX)

## Acknowledgments

The trial team wishes to express gratitude to the Jimma University, Maji woreda, and the participating mothers.

## Author Contributions

**Conceptualization:** Abraham Tamirat Gizaw, Pradeep Sopory, Morankar N. Sudhakar.

**Data curation:** Abraham Tamirat Gizaw, Pradeep Sopory.

**Formal analysis:** Abraham Tamirat Gizaw, Pradeep Sopory, Morankar N. Sudhakar.

**Funding acquisition:** Abraham Tamirat Gizaw.

**Investigation:** Abraham Tamirat Gizaw, Morankar N. Sudhakar.

**Methodology:** Abraham Tamirat Gizaw, Pradeep Sopory, Morankar N. Sudhakar.

**Project administration:** Abraham Tamirat Gizaw, Morankar N. Sudhakar.

**Resources:** Abraham Tamirat Gizaw.

**Software:** Abraham Tamirat Gizaw, Pradeep Sopory.

**Supervision:** Abraham Tamirat Gizaw, Morankar N. Sudhakar.

**Validation:** Abraham Tamirat Gizaw, Morankar N. Sudhakar.

**Visualization:** Abraham Tamirat Gizaw, Pradeep Sopory, Morankar N. Sudhakar.

**Writing – original draft:** Abraham Tamirat Gizaw.

**Writing – review & editing:** Pradeep Sopory, Morankar N. Sudhakar.

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
