## [Decision Letter · Decision Letter 0]

17 May 2022

PONE-D-22-07220Effectiveness of a positive deviant intervention to improve appropriate feeding practices and nutritional outcomes in West Omo Zone, Maji District: South West Region, Ethiopia: A study protocol for a cluster randomized control trialPLOS ONE

Dear Dr. Gizaw,

Thank you for submitting your manuscript to PLOS ONE. After careful consideration, we feel that it has merit but does not fully meet PLOS ONE’s publication criteria as it currently stands. Therefore, we invite you to submit a revised version of the manuscript that addresses the points raised during the review process.

The investigators deserve to be complimented for attempting a challenging study. However there are major concerns that have been raised by the reviewers regarding several important issues in the study design, sample size calculation, ethics, data analysis and conclusions. These will have to be satisfactorily addressed by the authors before the manuscript can be considered for publication. 

We look forward to receiving your revised manuscript.

Kind regards,

Sourabh Dutta

Academic Editor

PLOS ONE

Journal Requirements:

2. Thank you for submitting the above manuscript to PLOS ONE. During our internal evaluation of the manuscript, we found significant text overlap between your submission and the following previously published works, some of which you are an author.

-   https://link.springer.com/article/10.1186/s12887-018-1278-5

Please revise the manuscript to rephrase the duplicated text, cite your sources, and provide details as to how the current manuscript advances on previous work. Please note that further consideration is dependent on the submission of a manuscript that addresses these concerns about the overlap in text with published work.

5. We note you have included a table to which you do not refer in the text of your manuscript. Please ensure that you refer to Table 1 in your text; if accepted, production will need this reference to link the reader to the Table.

6. Please include a copy of Table 2 which you refer to in your text on page 9.

Additional Editor Comments:

The reviewers have asked for major changes in the manuscript. They have important concerns regarding the study design, sample size calculation, analysis and the conclusions drawn. The manuscript can be considered for publication if major revisions are made as requested by the reviewers, but the submission of a revised manuscript is not a guarantee for publication.

Reviewers' comments:

Reviewer's Responses to Questions

**Comments to the Author**

1. Does the manuscript provide a valid rationale for the proposed study, with clearly identified and justified research questions?

Reviewer #1: No

Reviewer #2: Yes

2. Is the protocol technically sound and planned in a manner that will lead to a meaningful outcome and allow testing the stated hypotheses?

Reviewer #1: Yes

Reviewer #2: Partly

3. Is the methodology feasible and described in sufficient detail to allow the work to be replicable?

Reviewer #1: Yes

Reviewer #2: No

4. Have the authors described where all data underlying the findings will be made available when the study is complete?

Reviewer #1: Yes

Reviewer #2: No

5. Is the manuscript presented in an intelligible fashion and written in standard English?

Reviewer #1: Yes

Reviewer #2: Yes

6. Review Comments to the Author

You may also provide optional suggestions and comments to authors that they might find helpful in planning their study.

Reviewer #1: Comments

The study protocol by Gizaw et al., is one of a kind which describes the effectiveness of a positive deviant intervention to improve appropriate feeding practices and nutritional outcomes in West Omo Zone, Maji District Ethiopia. But there are some revisions to be made in the paper, pertaining to which the manuscript would be suitable for publication:

Abstract:

Comment 1: The author used the word inclusion criteria in the abstract. A brief sentence can be added about how the author selected the paricipants based on inclusion and exclusion criteria.

Introduction:

Comment 2: Does the introduction provide sufficient background information for readers not in the immediate field to understand the problem/hypotheses?

The introduction provides a good, generalized background of the topic that quickly gives the reader an appreciation of the positive deviant to improve appropriate feeding practices and nutritional outcomes. However, to make the introduction more substantial, the author may wish to provide several references to substantiate the claim made in the second paragraph (that is, provide references to other groups who do or have done research in this area).

Methods

Comment 3: In the methods, a sentence could be added on the pilot survey after which the questionnaire and tools could be modified.

Comment 4: Before data collection among participants, how the reliability and validity of the translated questionnaire was checked?

Discussion

Comment 5: Limitations of the study could be added.

Reviewer #2: Review Comments to the Author

• I congratulate the authors for having decided to work in much needed field. The study design is appropriate for the problem identified, but needs some further work.

• Before getting into the specific comments of the study protocol per se, I would like to rise two concerns:

1. The Ethical Clearance body and the funding agency/ body seem to be the same.

2. The study lacks formative research such as KAP and Baseline data on the present feeding practice and nutritional outcomes. These are needed for an approach such as positive deviant study. These could be added in the methods.

• The abstract is well written and covers the main points of the study. The research question, the approach of the study, the intervention and the analysis are stated clearly. However, the authors mention using the experiences of the mothers who have come up with solutions to their IYCF problems. But later they say that they will use WHO IYCF guidelines. These are contradictory statements in the abstract.

• Introduction : The background information on the concept of Positive Deviance and the evidence of effectiveness of positive deviant nutritional education on the nutritional outcomes of the children are missing in the introduction. Need to be added.

• Health is often spelt as ‘hearth’

• The intervention describes the positive deviance/health nutrition education as an approach that identifies and transfers good behaviours shown by mothers of well-nourished children from disadvantaged households to others in the community with malnourished children. Therefore, the positive deviant mother should have a child with a similar age group to enable the positive deviant mother to show the effects of positive deviance. The study does not mention this as a criteria. Additionally, the protocol does not describe the method of assessing such as conducting anthropometry to assess growth and nutritional status of children, to select the successful mothers.

In the study protocol researchers have mentioned that the intervention is based on WHO Infant and young child feeding practices guidelines which will be developed by the researcher in the local language. The delivery of these guidelines is done by positive deviant mothers. Here mothers whose children are normal in nourishment are trained and made as health educators. [called as positive deviant mothers].

Practices used by such positive deviant mothers are not explored and used in correcting the malnourishment. Sustainability and locally followed ‘deviant’ and unique practices, which are the core principles of positive deviance. As this method is sustainable if such mothers are willing to act as the health educators, and the recipient mothers willing to accept their example.

Questions and concerns regarding the acceptance of the PD mothers as a motivation and authoritative educators need to be addressed. Measures to ensure the same is to be thought through. Moreover, which messages will the PD mother deliver ? Their own practices that were successful or the WHO guidelines. If the later, then this is not a PD method. It is only using successful mothers as the messengers.

• Eligibility criteria for cluster choosing, is vague. If the researcher is choosing non-adjacent, geographically accessible zone, it is a non-random selection of the zones purposively. Such a step may be necessary for hassle-free completion of the said study. However, it is not appropriate to mention such type of selection as random selection.

• As authors describe the protocol to be single blinded parallel cluster randomized trial, there has been no description on the methods used to ensure blinding and, at what stages blinding will be ensured.

• In the selection criteria of the participants, it is helpful to specify the conditions/diseases under severely ill [inclusion and exclusion criteria for severely ill].

• The age group of the study population is mentioned as 0-24 months. However, there are several components of the study like complimentary feeding, Minimum Meal Frequency, minimum dietary diversity cannot be assessed in the age group of 0-6 months owing to the period of exclusive breast feeding. A prior plan to address such issues, to have enough sample size for reliable results is needed.

• Sample Size estimation:

The p2 in the denominator for sample size calculation appears to be incorrect. There is a discrepancy in the value of the p2 mentioned and the one used in the formula. The number of clusters does not account for the total sample size that includes the additional loss to follow-up numbers. Therefore, it might be useful to check the math again.

The said sample size of 516 to be selected in each of the arms of the study. Sample size estimation formula is to provide the sample size for one arm. So, actual size should be 516 in the intervention arm and 516 in the control arm. So, the total sample will be 1032 if the said power of the study and reliable results is wished.

The intra cluster correlation coefficient of 0.03 seems low considering the nutritional and feeding practices within the said cluster remain similar.

• Sampling: Method of selection of the [non-positive deviant] women within the cluster is not mentioned i.e., How are 12 women in each of the clusters selected? What is the total expected number of mothers in each zone ? How the 12 will be selected from them ?

• Intervention

Although researcher has mentioned the selection criteria of the positive deviant mothers. However, how did the researcher identify the normal children of the selected positive deviant mothers? And what was the sampling techniques used to select the PD mothers amongst the mothers who have fulfilled the inclusion criteria mentioned?

The ratio of positive deviant mother to the selected non-positive deviant mothers is to be specified. The detailed session plan and the methods of delivery of health education sessions must be described including the time and place of demonstrations should be specified.

The outcome of change in the nutritional status of the child is being measured. But, How and who will be monitoring and/or measuring the behavioral change in the non-positive deviant mothers?

It is desired to mention the ratio of the supervisor to positive deviant mothers involved in the program and training and quality assurance of the work of the supervisors.

Issues related to protocol about the selection of mothers or positive deviant mothers must be addressed.

• Control

The researcher mentions that the control arm receives the standard care. It necessary to mention what comprises the standard care.

• In an RCT, there is need to show the baseline equivalence between the intervention and the control arm. i.e., that the baseline characteristics of both the arms are comparable and so, later the researcher can measure the effect of the intervention.

• Prior identification of possible bias and confounders can help in addressing such if present in the plan of analysis. Measures used in addressing the possibilities of Hawthorn effect in the control arm is not mentioned.

• Research hypothesis:

The research hypothesis mentioned that, there will be significant improvement. However, it is important the quantify the expected improvement.

The research hypothesis seems like a one-sided hypothesis with superiority trial. However, values of alpha and beta used in the sample size estimation and p value of the level of significance is taken of the two-sided hypothesis testing.

• Secondary objective: There is hardly any mention about the methodology viz. data collection methods or the qualitative methods to achieve secondary objectives.

• The study does not account for the possible programmatic issues that may arise in the even the positive deviant mother’s child falls ill and the solutions/programmatic implications it may have. Crisis management: If there are any children [who are participants] during the study period who fall severely ill or decrease in the weight becoming severely malnourished. What is the plan of action in such circumstances: in terms of care and treatment of such children and their continuation as participants in the study. Care should be taken to address growth faltering

• The ‘routene interventions’ – will they be continued in the intervention arm, as in the control arm ?

• Mention of the tools used for assessing complimentary feeding, description of such data and their plan analysis is needed.

• Mid Arm Circumference measurement is mentioned in the data collection. Please ensure that the specifics mentioned in the Table 2 match the description and are the same throughout the various sections of the paper. Although, the methods to be used for the measurement such as Shikar’s tape, Bangal test or Measuring with tape etc., is not described. Use of the same in achieving the objective and plan of analysis is to be mentioned.

• In the table of assessment, there is mention of diarrhea, cough and fever. However, throughout the methodology there is no mention of using clinical assessment of the children its accountability of the outcome assessment.

• In the table of assessment, mention of household food insecurity assessment, water, hygiene and sanitation, cultural food taboos are present. However, no description such variable in the objectives, tools of assessment and data analysis is present in the protocol.

• Plan of Analysis: Whenever we have two comparison in survival analysis [intervention and control arms], it appropriate to use Cox’s proportional hazard model for analysis instead of Kaplan Meir curve.

• More detailed description of the plan of analysis in comparison of the intervention and control group at three different time lines is desirable.

• References needs to be revisited in the view of using original references to quote the data. For example, the authors have referred to studies from Cambodia, Ethiopia for WHO recommendations (references 2 and 3). It would be useful to refer to the original WHO documents. Similarly, while quoting global figures, it would be useful to quote the latest available data rather than studies from Maharashtra, which is perhaps not the original source of global data on malnutrition. Also, the local data from the study area should constitute the background/baseline. These are missing.

7. PLOS authors have the option to publish the peer review history of their article (what does this mean?). If published, this will include your full peer review and any attached files.

Reviewer #1: **Yes: **Dr Mona Duggal

Reviewer #2: No

---

## [Author Response · Author response to Decision Letter 0]

14 Aug 2022

Dr. Sourabh Dutta, PLOS ONE 

Academic Editor

The effectiveness of a positive deviance approach to improve appropriate feeding and nutritional outcomes in South West Region, Ethiopia: A study protocol for a cluster randomized control trial: PONE-D-22-07220

We thank the Senior Editor as well as the reviewers for their comments and appreciate the opportunity to submit a revised manuscript. The manuscript has been revised in accordance with the Editor’s comments and suggestions. Our specific response to each comment is detailed below, including all of the corresponding revisions in the manuscript and supplement (pages and paragraphs correspond to the “All Markup” format in Microsoft Word).

Editor Comments Response Page Number

Please ensure that your manuscript meets PLOS ONE's style requirements, including those for file naming. We have tried to follow the PLOS ONE’s style N/A

During our internal evaluation of the manuscript, we found significant text overlap between your submission and the following previously published works, some of which you are an author. We have made significant changes on the paragrahs and senteces that overlapped with the previous publications N/A

We note that the grant information you provided in the ‘Funding Information’ and ‘Financial Disclosure’ sections do not match. We have admitted the errors and now it is amended in the manuscript. Actually it was expected to be funded by Jimma Uniiversity and due to different budget constraints the University unable to fund us the grant informations were remove from the document and only we have secured ethical letter from Jimma university. N/A 

Please include your full ethics statement in the ‘Methods’ section of your manuscript file. In your statement, please include the full name of the IRB or ethics committee who approved or waived your study, as well as whether or not you obtained informed written or verbal consent. If consent was waived for your study, please include this information in your statement as well. We have written full name of the Ethical review board name and details Page 18

We note you have included a table to which you do not refer in the text of your manuscript. Please ensure that you refer to Table 1 in your text; if accepted, production will need this reference to link the reader to the Table. We have addressed and refered table 1. Page 12

Please include a copy of Table 2 which you refer to in your text on page 9. We have admitted that it was mistakenly written and there is no table 2 and we have corrected on the manuscript. N/A 

Please include captions for your Supporting Information files at the end of your manuscript, and update any in-text citations to match accordingly. Please see our Supporting Information guidelines for more information: http://journals.plos.org/plosone/s/supporting-information.

We have accepted the comment and well addressed Page 20

Reviewer 1 comments 

Comment 1: The author used the word inclusion criteria in the abstract. A brief sentence can be added about how the author selected the paricipants based on inclusion and exclusion criteria. We have accepted the comment and due to the limitation of word count in the abstract, we have removed the term “ inclusion criteria” from the abstract. However, we have added the detailed description of inclusion and exclusion criteria under the subheading eligibility Page 6

Comment 2: Does the introduction provide sufficient background information for readers not in the immediate field to understand the problem/hypotheses?

The introduction provides a good, generalized background of the topic that quickly gives the reader an appreciation of the positive deviant to improve appropriate feeding practices and nutritional outcomes. However, to make the introduction more substantial, the author may wish to provide several references to substantiate the claim made in the second paragraph (that is, provide references to other groups who do or have done research in this area). We have addressed the comment in the second paragraph Page 3

Comment 3: In the methods, a sentence could be added on the pilot survey after which the questionnaire and tools could be modified. We have added the sentences under the subheading “Data quality assurance” Page 18

Comment 4: Before data collection among participants, how the reliability and validity of the translated questionnaire was checked? We have addressed the comment under the subheading “Data quality assurance” Page 18

Comment 5: Limitations of the study could be added. We have added the limitation of the study under the subheading “discussion” Page 19 and 20

Reviewer 2 comments 

The Ethical Clearance body and the funding agency/ body seem to be the same. We have initially thaught the this project will be funded by Jimma University, However by now we have only secured ethical clearance from Jimma University. N/A

 The study lacks formative research such as KAP and Baseline data on the present feeding practice and nutritional outcomes. These are needed for an approach such as positive deviant study. These could be added in the methods. We have added under the subheading “design” Page 5

The abstract is well written and covers the main points of the study. The research question, the approach of the study, the intervention and the analysis are stated clearly. However, the authors mention using the experiences of the mothers who have come up with solutions to their IYCF problems. But later they say that they will use WHO IYCF guidelines. These are contradictory statements in the abstract. We have addressed theh comment in the abstract. However we need to also provide some clues regarding the reviewers comment. The mothers have practical ways of doing uncommon behaviour that is important for their child. However, for supporting non-positive deviant mothers we need to support and equip them with the knowledge which is recommende by WHO, which is not in contrary to the uncommon practice rather in building their knowledge for facilitating change to mothers in the intervention group. The practical demonistration is exceptional addressed through thrier experiences built by their own for theirown problems. Page 2 (as highlight)

Page 8-9, described in details.

Introduction : The background information on the concept of Positive Deviance and the evidence of effectiveness of positive deviant nutritional education on the nutritional outcomes of the children are missing in the introduction. Need to be added. We have added in the introduction. Page 4 and 5

Health is often spelt as ‘hearth’ We have corrected it and actually it is not to mean as health. Some literature uses the term interchangeably like positive deviance/ hearth. Here are few evidences : https://coregroup.org/wp-content/uploads/2017/09/Positive-Deviance-Hearth-Resource-Guide.pdf

However, for the mutual understanding not to confuse the reader after publication we decided to use only the term positive deviance approach. N/A

The intervention describes the positive deviance/health nutrition education as an approach that identifies and transfers good behaviours shown by mothers of well-nourished children from disadvantaged households to others in the community with malnourished children. Therefore, the positive deviant mother should have a child with a similar age group to enable the positive deviant mother to show the effects of positive deviance. The study does not mention this as a criteria. Additionally, the protocol does not describe the method of assessing such as conducting anthropometry to assess growth and nutritional status of children, to select the successful mothers. We have addressed the constructive comments provided by the reviewer and we admit that the comments are important. Page 8-10

• The intervention describes the positive deviance/health nutrition education as an approach that identifies and transfers good behaviours shown by mothers of well-nourished children from disadvantaged households to others in the community with malnourished children. Therefore, the positive deviant mother should have a child with a similar age group to enable the positive deviant mother to show the effects of positive deviance. The study does not mention this as a criteria. Additionally, the protocol does not describe the method of assessing such as conducting anthropometry to assess growth and nutritional status of children, to select the successful mothers.

In the study protocol researchers have mentioned that the intervention is based on WHO Infant and young child feeding practices guidelines which will be developed by the researcher in the local language. The delivery of these guidelines is done by positive deviant mothers. Here mothers whose children are normal in nourishment are trained and made as health educators. [called as positive deviant mothers].

Practices used by such positive deviant mothers are not explored and used in correcting the malnourishment. Sustainability and locally followed ‘deviant’ and unique practices, which are the core principles of positive deviance. As this method is sustainable if such mothers are willing to act as the health educators, and the recipient mothers willing to accept their example.

Questions and concerns regarding the acceptance of the PD mothers as a motivation and authoritative educators need to be addressed. Measures to ensure the same is to be thought through. Moreover, which messages will the PD mother deliver ? Their own practices that were successful or the WHO guidelines. If the later, then this is not a PD method. It is only using successful mothers as the messengers. We have addressed the concern raised by the reviewer in different part in the manuscript. 1. Page 8-10 details were included (positive deviant issues)

2. Page 15

(anthropometric measurement)

Eligibility criteria for cluster choosing, is vague. If the researcher is choosing non-adjacent, geographically accessible zone, it is a non-random selection of the zones purposively. Such a step may be necessary for hassle-free completion of the said study. However, it is not appropriate to mention such type of selection as random selection. We have accepted and addressed the comment. Page 6

As authors describe the protocol to be single blinded parallel cluster randomized trial, there has been no description on the methods used to ensure blinding and, at what stages blinding will be ensured. We have added this to the method part. Page 7

In the selection criteria of the participants, it is helpful to specify the conditions/diseases under severely ill [inclusion and exclusion criteria for severely ill]. We have addressed the comment and added it in the method part. Page 6

The age group of the study population is mentioned as 0-24 months. However, there are several components of the study like complimentary feeding, Minimum Meal Frequency, minimum dietary diversity cannot be assessed in the age group of 0-6 months owing to the period of exclusive breast feeding. A prior plan to address such issues, to have enough sample size for reliable results is needed. We appreciate the views of the reviewer and actually we have modified the scope of the study to address the cognitive domain and affective domain of the mothers such as knowledge, attitude and self-efficacy of mmothers and it is not the practice. As mentioned above 0-6 months mothers will practice exclusive breastfeeding, but after few months the mothers will proceed to complementary feeding. Therefore, the knowledge, attitude and complementary feeding assessed simultaneously. Page 14

• Sample Size estimation:

The p2 in the denominator for sample size calculation appears to be incorrect. There is a discrepancy in the value of the p2 mentioned and the one used in the formula. The number of clusters does not account for the total sample size that includes the additional loss to follow-up numbers. We have addressed the method part of the manuscript. Page 7

Sampling: Method of selection of the [non-positive deviant] women within the cluster is not mentioned i.e., How are 12 women in each of the clusters selected? What is the total expected number of mothers in each zone ? How the 12 will be selected from them ? We have addressed in the method part Page 6

Intervention

Although researcher has mentioned the selection criteria of the positive deviant mothers. However, how did the researcher identify the normal children of the selected positive deviant mothers? And what was the sampling techniques used to select the PD mothers amongst the mothers who have fulfilled the inclusion criteria mentioned?

The ratio of positive deviant mother to the selected non-positive deviant mothers is to be specified. The detailed session plan and the methods of delivery of health education sessions must be described including the time and place of demonstrations should be specified.

The outcome of change in the nutritional status of the child is being measured. But, How and who will be monitoring and/or measuring the behavioral change in the non-positive deviant mothers?

It is desired to mention the ratio of the supervisor to positive deviant mothers involved in the program and training and quality assurance of the work of the supervisors.

Issues related to protocol about the selection of mothers or positive deviant mothers must be addressed. We have addressed the details in the method part Page 8-10

The researcher mentions that the control arm receives the standard care. It necessary to mention what comprises the standard care. We have addressed in the method part Page 11

In an RCT, there is need to show the baseline equivalence between the intervention and the control arm. i.e., that the baseline characteristics of both the arms are comparable and so, later the researcher can measure the effect of the intervention. Prior identification of possible bias and confounders can help in addressing such if present in the plan of analysis. Measures used in addressing the possibilities of Hawthorn effect in the control arm is not mentioned.

 We have addressed in the method part. Through blinding that we can address the howthorn effect. Page 5 and 18

• Research hypothesis:

The research hypothesis mentioned that, there will be significant improvement. However, it is important the quantify the expected improvement.

The research hypothesis seems like a one-sided hypothesis with superiority trial. However, values of alpha and beta used in the sample size estimation and p value of the level of significance is taken of the two-sided hypothesis testing.

• Secondary objective: There is hardly any mention about the methodology viz. data collection methods or the qualitative methods to achieve secondary objectives We have addressed in method part Page 12

The study does not account for the possible programmatic issues that may arise in the even the positive deviant mother’s child falls ill and the solutions/programmatic implications it may have. Crisis management: If there are any children [who are participants] during the study period who fall severely ill or decrease in the weight becoming severely malnourished. What is the plan of action in such circumstances: in terms of care and treatment of such children and their continuation as participants in the study. Care should be taken to address growth faltering We have addressed the method part Page 9

The ‘routene interventions’ – will they be continued in the intervention arm, as in the control arm ? We have addressed it in the method part. For information the routine services provided by health extension workers are provided without any interferences. Page 11

Mention of the tools used for assessing complimentary feeding, description of such data and their plan analysis is needed. We have addressed in the method part Page 16 

Mid Arm Circumference measurement is mentioned in the data collection. Please ensure that the specifics mentioned in the Table 2 match the description and are the same throughout the various sections of the paper. Although, the methods to be used for the measurement such as Shikar’s tape, Bangal test or Measuring with tape etc., is not described. Use of the same in achieving the objective and plan of analysis is to be mentioned. We have accepted the comment and modified it accordingly. For instance the the mentioned table 2 is mistakenly written and it is Table 1, we have removed MUAC and included details of anthropometric measurement in the method part of the manuscript. Page 16

In the table of assessment, there is mention of diarrhea, cough and fever. However, throughout the methodology there is no mention of using clinical assessment of the children its accountability of the outcome assessment. We have added to under different subheading of the method part. Page 13 and 16

In the table of assessment, mention of household food insecurity assessment, water, hygiene and sanitation, cultural food taboos are present. However, no description such variable in the objectives, tools of assessment and data analysis is present in the protocol. We have addressed under different subheading in the method. Due to the presence of some data on food insecurity,and sanitation and hygiene we have removed it from the manuscript. But the child morbidity and taboos are mentioned in method part of the manuscript. Page 5,13 and 16

Plan of Analysis: Whenever we have two comparison in survival analysis [intervention and control arms], it appropriate to use Cox’s proportional hazard model for analysis instead of Kaplan Meir curve We have added in the method part Page 17

More detailed description of the plan of analysis in comparison of the intervention and control group at three different time lines is desirable. We have added in the method part Page 17

References needs to be revisited in the view of using original references to quote the data. For example, the authors have referred to studies from Cambodia, Ethiopia for WHO recommendations (references 2 and 3). It would be useful to refer to the original WHO documents. Similarly, while quoting global figures, it would be useful to quote the latest available data rather than studies from Maharashtra, which is perhaps not the original source of global data on malnutrition. Also, the local data from the study area should constitute the background/baseline. These are missing. We have added in the reference part Page 20 and 21

Thank you for the invaluable comments!

Dr. Sourabh Dutta, PLOS ONE Date: 14/08/2022

Academic Editor

Effectiveness of a positive deviance approach to improve appropriate feeding and nutritional outcomes in South West Region, Ethiopia: A study protocol for a cluster randomized control trial: PONE-D-22-07220

We thank the Senior Editor as well as the reviewers for their comments and appreciate the opportunity to submit a revised manuscript. The manuscript has been revised in accordance with the Editor’s comments and suggestions. Our specific response to each comment is detailed below, including all of the corresponding revisions in the manuscript and supplement (pages and paragraphs correspond to the “All Markup” format in Microsoft Word).

Editor Comments Response Page Number

1. We notice that your manuscript file was uploaded on March 14, 2022. Please can you upload the latest version of your revised manuscript as the main article file, ensuring that does not contain any tracked changes or highlighting. This will be used in the production process if your manuscript is accepted. Please follow this link for more information: http://blogs.PLOS.org/everyone/2011/05/10/how-to-submit-your-revised-manuscript/

We have separated the main manuscript without changes in the file name of 

“ATG-Main Manuscript revised Trial A1922” N/A 

2. Thank you for submitting the above manuscript to PLOS ONE. During our internal evaluation of the manuscript, we found significant text overlap between your submission and the following previously published works, some of which you are an author.

- https://link.springer.com/article/10.1186/s12887-018-1278-5

Please revise the manuscript to rephrase the duplicated text, cite your sources, and provide details as to how the current manuscript advances on previous work. Please note that further consideration is dependent on the submission of a manuscript that addresses these concerns about the overlap in text with published work.

 We have made all the significant changes, definitely paraphrased all the similar phrases and paragraphs. N/A

We note that the grant information you provided in the ‘Funding Information’ and ‘Financial Disclosure’ sections do not match. When you resubmit, please ensure that you provide the correct grant numbers for the awards you received for your study in the ‘Funding Information’ section. We request excuse for the fund number and information provided. The grant that we perceived to be secured from Jimma University, is not yet secured. Therefore, there is no funding information that we can provide. We have secured only ethical letter from Jimma University. N/A

Your ethics statement should only appear in the Methods section of your manuscript. If your ethics statement is written in any section besides the Methods, please move it to the Methods section and delete it from any other section. Please ensure that your ethics statement is included in your manuscript, as the ethics statement entered into the online submission form will not be published alongside your manuscript. We have added it to the method part. Page 17

 Please ensure that you refer to Figure 3 in your text as, if accepted, production will need this reference to link the reader to the figure We have added it wrongly and removed it from the lists of figures. N/A

---

## [Decision Letter · Decision Letter 1]

30 Aug 2023

PONE-D-22-07220R1Effectiveness of a positive deviance approach to improve appropriate feeding and nutritional outcomes in South West Region, Ethiopia: A study protocol for a cluster randomized control trialPLOS ONE

Dear Dr. Gizaw,

Thank you for submitting your manuscript to PLOS ONE. After careful consideration, we feel that it has merit but does not fully meet PLOS ONE’s publication criteria as it currently stands. Therefore, we invite you to submit a revised version of the manuscript that addresses the points raised during the review process.

We look forward to receiving your revised manuscript.

Kind regards,

Jianhong Zhou

Staff Editor

PLOS ONE

Journal Requirements:

Additional Editor Comments:

As reviewer 2 and reviewer 3 have hinted, the manuscript is rather confusing to follow. There are fundamental problems in understanding what exactly is the definition of "a positive deviant mother" . 

As per page 6, the NON-positive deviant mothers should have a infant/child aged 0-24 months, the child's Height-for-age Z (HAZ) scores HAZ < −2 and a child with NO severe malnutrition will be included in the study. Presumably <-2 z score means worse than -2. It is not clear from "a child with no severe malnutrition" whether this refers to the index child who is expected to be stunted or a sibling of the child. If it is the index child, then is the child supposed to have no severe malnutrition but be stunted at the same time, and if so, why?

On page 9, the POSITIVE deviant mothers are defined as those who have a number of positive attributes but their child should apparently have the following attributes which are all negative "was a big baby who is losing weight now", "have any severely malnourished sibling", "have any serious or typical social or health problem", "have families enrolled in a supplementary feeding program".  

Apart from the ambiguity about the definition of positive deviant and nonpositive deviant mothers, the manuscript lacks clarity in many places. 

The authors must take the help of a colleague whose primary language is English. There are many sentences in the manuscript that are challenging to understand. Perhaps the authors were not able to get their point across clearly because of a language barrier.

Reviewers' comments:

Reviewer's Responses to Questions

**Comments to the Author**

1. Does the manuscript provide a valid rationale for the proposed study, with clearly identified and justified research questions?

Reviewer #1: Yes

Reviewer #3: Yes

2. Is the protocol technically sound and planned in a manner that will lead to a meaningful outcome and allow testing the stated hypotheses?

Reviewer #1: Yes

Reviewer #3: Partly

3. Is the methodology feasible and described in sufficient detail to allow the work to be replicable?

Reviewer #1: Yes

Reviewer #3: Yes

4. Have the authors described where all data underlying the findings will be made available when the study is complete?

Reviewer #1: Yes

Reviewer #3: Yes

5. Is the manuscript presented in an intelligible fashion and written in standard English?

Reviewer #1: No

Reviewer #3: Yes

6. Review Comments to the Author

You may also provide optional suggestions and comments to authors that they might find helpful in planning their study.

Reviewer #1: The previous reviewer's comments have been addressed . The methodology has been strengthened and strengths and limitations of the stud have been added.

Reviewer #3: In this Randomized control trial, authors are trying to identify positive deviant mothers , utilise them to educate other mothers in the community whose children are stunted and then to study the effectiveness of the approach to improve feeding and nutritional outcomes in Ethiopia. This is a study addressed to improve a very important issue of the nation that is malnutrition.

Comment 1: The study design is stated to be a single blinded RCT . It is stated that because of the nature of intervention, mothers cannot be blinded but the intervention assignment is concealed from the interviewers collecting outcome data. But how will it be possible to conceal the intervention data from the interviewers as they are collecting data from mother herself who knows it? How is blinding possible in such a situation?

Comment 2: Regarding identification of positive deviant mother , the criteria from child’s side is stated to be a big baby who is losing weight now , have any severely malnourished sibling , have any social or health problem are taken as babies who will be considered. But should it not be other way round ? Their families should not be considered for positive deviance.

Comment 3: Though the authors tried to mention that every fortnightly positive deviant mothers will visit non positive deviant mothers, the practical aspects like how the they would be transported , the logistics , time taken to do so , how they are going to help other mothers need to be addressed.

7. PLOS authors have the option to publish the peer review history of their article (what does this mean?). If published, this will include your full peer review and any attached files.

Reviewer #1: **Yes: **Mona Duggal

Reviewer #3: No

---

## [Author Response · Author response to Decision Letter 1]

5 Sep 2023

To: Dr. Jianhong Zhou

Staff Editor

PLOS ONE

Title of the manuscript : " Effectiveness of a positive deviance approach to improve appropriate feeding and nutritional status in South West Region, Ethiopia: A study protocol for a cluster randomized control trial"

We thank the senior editor for their comments and appreciate the opportunity to submit a revised manuscript. The required documents were attached in accordance with the comments and suggestions. 

Comment of the editor Response Page Number

Please review your reference list to ensure that it is complete and correct. If you have cited papers that have been retracted, please include the rationale for doing so in the manuscript text, or remove these references and replace them with relevant current references. Any changes to the reference list should be mentioned in the rebuttal letter that accompanies your revised manuscript. If you need to cite a retracted article, indicate the article’s retracted status in the References list and also include a citation and full reference for the retraction notice. We have addressed the comment on the manuscript. We have excluded the retracted references, corrected cross references and used updated possible as much as possible. However, some references are very important which depicted the steps of conducting positive deviant approach and we haven’t found updated references and we used it as it is. 26,27 & 28

As reviewer 2 and reviewer 3 have hinted, the manuscript is rather confusing to follow. There are fundamental problems in understanding what exactly is the definition of "a positive deviant mother”. 

 We have included in the method section of the manuscript 11

As per page 6, the NON-positive deviant mothers should have a infant/child aged 0-24 months, the child's Height-for-age Z (HAZ) scores HAZ < −2 and a child with NO severe malnutrition will be included in the study. Presumably <-2 z score means worse than -2. It is not clear from "a child with no severe malnutrition" whether this refers to the index child who is expected to be stunted or a sibling of the child. If it is the index child, then is the child supposed to have no severe malnutrition but be stunted at the same time, and if so, why? We appreciate the comment that the editor trying to raise. It is quite confusing and we have addressed the comment in the “eligibility criteria for the participants” section of the manuscript. 

 8

On page 9, the POSITIVE deviant mothers are defined as those who have a number of positive attributes but their child should apparently have the following attributes which are all negative "was a big baby who is losing weight now", "have any severely malnourished sibling", "have any serious or typical social or health problem", "have families enrolled in a supplementary feeding program". 

 We have addressed the comment in the “intervention” section of the manuscript. We totally edited the section. 11

Apart from the ambiguity about the definition of positive deviant and nonpositive deviant mothers, the manuscript lacks clarity in many places. 

 We have addressed the comment in the “intervention” section of the manuscript. We tried to maintain clarity throughout the manuscript by editing the language errors and grammar errors. 11

The authors must take the help of a colleague whose primary language is English. There are many sentences in the manuscript that are challenging to understand. Perhaps the authors were not able to get their point across clearly because of a language barrier.

 In addition to the corresponding authors, one of the co-author from USA took full responsibility in editing the language throughout the manuscript . N/A

Reviewer #1: The previous reviewer's comments have been addressed. The methodology has been strengthened and strengths and limitations of the stud have been added. No comments are provided and we appreciate for being acting as a reviewer of our manuscript. N/A

Reviewer 3 comments 

Comment 1: The study design is stated to be a single blinded RCT . It is stated that because of the nature of intervention, mothers cannot be blinded but the intervention assignment is concealed from the interviewers collecting outcome data. But how will it be possible to conceal the intervention data from the interviewers as they are collecting data from mother herself who knows it? How is blinding possible in such a situation?

 Dear reviewer thanks for raising most important concern. However, concealing the intervention group from the interviewer are most important for assuring and understanding the effect of the intervention (PDA). This potentially minimize the bias of the data collection. This can be done by recruiting the data collectors who are away or living out of the intervention area. The baseline, midline and endline data collection will be conducted simultaneously both to the intervention and control groups, where the PI and Co-PI know, but the data collectors don’t know. The cluster will be coded (01-36) and hh-labelling (001-516) will be given. The research team knows which cluster is the intervention and control group at the end of the data collection. Also the study hypothesis also concealed from health extension workers and other volunteers working on health because once they know it they will bias the outcome by providing addition lesson on IYCFP to routine health education. This will bias the effect of the PDA. 9

Comment 2: Regarding identification of positive deviant mother , the criteria from child’s side is stated to be a big baby who is losing weight now , have any severely malnourished sibling , have any social or health problem are taken as babies who will be considered. But should it not be other way round? Their families should not be considered for positive deviance. We have addressed the comment in the “intervention” section of the manuscript. We totally amended the section. 11

Comment 3: Though the authors tried to mention that every fortnightly positive deviant mothers will visit non positive deviant mothers, the practical aspects like how the they would be transported , the logistics , time taken to do so , how they are going to help other mothers need to be addressed. We have addressed the comment in the “Intervention activities” section of the manuscript. We totally amended the section. 13 & 14

Thank you for the important comments!

---

## [Decision Letter · Decision Letter 2]

24 Oct 2023

Effectiveness of a positive deviance approach to improve appropriate feeding and nutritional status in South West Region, Ethiopia: A study protocol for a cluster randomized control trial

PONE-D-22-07220R2

Dear Dr. Gizaw,

We’re pleased to inform you that your manuscript has been judged scientifically suitable for publication and will be formally accepted for publication once it meets all outstanding technical requirements.

Kind regards,

Sourabh Dutta

Academic Editor

PLOS ONE

Additional Editor Comments (optional):

The manuscript is acceptable for publication. Although one of the reviewers has asked for a minor revision (that the eligibility criteria of the positive deviant mothers has to be mentioned), this is not necessary because the authors have laid down the process of identifying the positive deviant mothers in detail as a part of the 6 steps on page 11 of the manuscript.

Reviewers' comments:

Reviewer's Responses to Questions

**Comments to the Author**

1. Does the manuscript provide a valid rationale for the proposed study, with clearly identified and justified research questions?

Reviewer #3: Yes

Reviewer #4: Yes

2. Is the protocol technically sound and planned in a manner that will lead to a meaningful outcome and allow testing the stated hypotheses?

Reviewer #3: Yes

Reviewer #4: Yes

3. Is the methodology feasible and described in sufficient detail to allow the work to be replicable?

Reviewer #3: Yes

Reviewer #4: Yes

4. Have the authors described where all data underlying the findings will be made available when the study is complete?

Reviewer #3: Yes

Reviewer #4: Yes

5. Is the manuscript presented in an intelligible fashion and written in standard English?

Reviewer #3: Yes

Reviewer #4: Yes

6. Review Comments to the Author

You may also provide optional suggestions and comments to authors that they might find helpful in planning their study.

Reviewer #3: The protocol is now comprehensible.It would be clear if the eligibility criteria contains information of all mothers to be included , not only non positive deviant mothers.

Reviewer #4: The manuscript is the study protocol of a cluster RCT aimed to assess the effectiveness of a positive deviance approach (PDA) to improve IYCF and nutritional status in South West region, Ethiopia. This is the second revision. Other than being very lengthy the protocol is well written

The study objectives are clear.

The rationale for the study is well described

The population, intervention and ouctomes are described well

The outcomes are defined and the analysis plan is presented

Sample size calculation is provided

The intervention – the positive deviant behaviours need to be identified and chosen to be delivered to the intervention clusters. The steps for this are provided in the protocol

Trial registration and ethical approvals are also provided.

7. PLOS authors have the option to publish the peer review history of their article (what does this mean?). If published, this will include your full peer review and any attached files.

Reviewer #3: **Yes: **Mandula Phani Priya

Reviewer #4: **Yes: **Sindhu Sivanandan

---

## [Editor Report · Acceptance letter]

22 Dec 2023

PONE-D-22-07220R2 

PLOS ONE

Dear Dr. Gizaw, 

I'm pleased to inform you that your manuscript has been deemed suitable for publication in PLOS ONE. Congratulations! Your manuscript is now being handed over to our production team.

Kind regards, 

on behalf of

Dr. Sourabh Dutta 

Academic Editor

PLOS ONE